# Neurocognitive Mechanism of Human Resilience: A Conceptual Framework and Empirical Review

**DOI:** 10.3390/ijerph16245123

**Published:** 2019-12-15

**Authors:** Zai-Fu Yao, Shulan Hsieh

**Affiliations:** 1Brain and Cognition, Department of Psychology, University of Amsterdam, 1018 WS Amsterdam, The Netherlands; zaifuyao@gmail.com; 2Cognitive Electrophysiology Laboratory: Control, Aging, Sleep, & Emotion (CASE), National Cheng Kung University, Tainan 701, Taiwan; 3Department of Psychology, College of Social Sciences, National Cheng Kung University, Tainan 701, Taiwan; 4Institute of Allied Health Sciences, College of Medicine, National Cheng Kung University, Tainan 701, Taiwan; 5Department and Institute of Public Health, College of Medicine, National Cheng Kung University, Tainan 701, Taiwan

**Keywords:** resilience, adversity, stress, cognitive flexibility, adaptive behavior

## Abstract

Resilience is an innate human capacity that holds the key to uncovering why some people rebound after trauma and others never recover. Various theories have debated the mechanisms underlying resilience at the psychological level but have not yet incorporated neurocognitive concepts/findings. In this paper, we put forward the idea that cognitive flexibility moderates how well people adapt to adverse experiences, by shifting attention resources between cognition–emotion regulation and pain perception. We begin with a consensus on definitions and highlight the role of cognitive appraisals in mediating this process. Shared concepts among appraisal theories suggest that cognition–emotion, as well as pain perception, are cognitive mechanisms that underlie how people respond to adversity. Frontal brain circuitry sub-serves control of cognition and emotion, connecting the experience of physical pain. This suggests a substantial overlap between these phenomena. Empirical studies from brain imaging support this notion. We end with a discussion of how the role of the frontal brain network in regulating human resilience, including how the frontal brain network interacts with cognition–emotion–pain perception, can account for cognitive theories and why cognitive flexibilities’ role in these processes can create practical applications, analogous to the resilience process, for the recovery of neural plasticity.

## 1. Introduction

Policy from the World Health Organization (WHO, Geneva, Switzerland) views resilience as a process that embraces positive adaptation, with protective factors and assets that moderate risk factors and therefore reduce the impact of risk on outcomes [1]. Likewise, the American Psychological Association (APA, Washington, DC, USA) defines resilience as ‘‘the process of adapting well in the face of adversity, trauma, tragedy, threats or even significant sources of stress’’ [2]. The importance of human resilience has emerged as a new frontier for studying the effects of adverse experiences on mental health and well-being [3,4]; nevertheless, its underlying mechanisms remain largely unknown [3,5]. Here, we aim to provide an initial step toward this deconstruction and interrogation. In particular, we highlight the important contributions from brain-imaging studies and explain the facets underlying this developmental process. We seek to provide a neurocognitive explanation of why people withstand and even thrive later in the face of adversity. In this paper, we first review the existing literature on resilience, defining resilience as a complex, high-level multidimensional human construct. We further summarize diverse works on resilience, outlining common feature across these definitions, possible mechanisms, and implications of relevant findings from classical experimental studies; subsequently, we propose a conceptual framework for understanding resilience. In the following section, we review the common scientific consensus on the factors relating to psychological resilience, for a better understanding of this process.

## 2. Resilience as a Dynamic Developmental Process

Most definitions of resilience are based around two core concepts, adversity and positive adaptation [6,7,8,9,10], which have to agree and are included in our integrative conceptual model (Figure 1). In Figure 1, resilience is defined as a dynamic developmental process to adapt and overcome negative outcomes over time, and we argue that the contents of the adverse experience, appraisal process, and positive adaptation are particularly critical to include in the definition, to build an integrative conceptual model for further examination by brain imaging methodology. In particular, the role of cognitive appraisal in the resilience process is understood as a moderator role to mitigate the negative effects of adverse experiences on adaptation.

For instance, within child and adolescent research, the achievement of salient developmental tasks in the face of adversity, such as learning to read and write, as well as attending and behaving properly at school, are viewed as positive adaptations/outcomes [7,11]. The period between beginning to learn and eventually being able to read and write may be considered a dynamic process of resilience. In line with these descriptions, an example from another stressful life event might also support this notion. Furthermore, bereaved persons who demonstrate the usual recovery pattern may exhibit symptoms of depression and experience difficulties completing their normal tasks at work, but they persevere and eventually begin to return to their pre-loss level of functioning over a period of one or two years [12]. The period between experiencing difficulties completing normal tasks at work and beginning to return to their pre-loss level of functioning that enables them to complete tasks at work may be seen as a result of positive adaption. Therefore, individuals who exhibit higher levels of resilience seem to be able to proceed with their lives with a minimal period of recovery or no apparent disruptions in their daily functioning.

Some of the conceptual difficulties around resilience are determined by the criteria that researchers use to assess whether the outcome is a ‘good’ one and reflects adaptation. Assessing resilience thus becomes a challenge. Despite the development of many psychometric measures to evaluate psychological resilience, its validity is still uncertain [13,14,15]. These measures have some missing information regarding the psychometric properties of resilience, suggesting a more rigorous theoretical basis to justify the results is warranted. Ideally, measures of resilience should be able to reflect the complexity of the concept and the temporal dimension. Adapting to change is a dynamic process. Neglecting the main antecedent may undermine understanding of this process. The collective importance of these notions is that they draw attention to questions about the nature or types of adverse experience, as well as the possibility of additive or cumulative effects on this adversity. In the next section, we explore the role of adversity in understanding resilience.

Our conceptual framework starts with the two foundations that make resilience a dynamic developmental process (Figure 1). In particular, the main antecedent of resilience is deemed to be adversity, and the aftermath is considered to be positive adaptation. The decisive role of cognitive appraisal in the resilience process is understood as its moderator role to mitigate the negative effects of adverse experiences on adaptation. There is a great variation in the human response to adverse experiences. In our view, a positive adaptive function can be modulated by cognitive appraisal and severity, unexpectedness (disruption of the predictive machinery in the generative inner model of the brain), and the subjective degree of physical pain’s apperception/painfulness (i.e., physical pain as a response to an emotion-regulation strategy [16]) of adverse experiences that would vary the brain mechanisms from top-down (e.g., goal-directed) cognitive control and bottom-up stimuli-driven processing. Bottom-up (e.g., stimuli-driven) processing refers to processing (e.g., goal-directed) that is built up from the passive, perceived external sensory information. Top-down processing, on the other hand, refers to active inference (e.g., high-level cognition) that is driven by prediction. From the attention–control theory perspective [17], anxiety impairs goal-directed attention and increases stimulus-driven attention. A similar mechanism is applicable to disproportionate levels of an adverse experiences; when facing an adverse situation, anxiety impairs top-down cognitive control/emotional regulation. Thus, people cannot perform flexible switching between negative–positive thoughts—they rely on more bottom-up, stimuli-driven attention that focuses on physical-pain processing.

## 3. Cognitive Appraisal of Adversity

### 3.1. The Perspective of Stressor Perception

The nature of the adversity can guide the strength of resilience, including severe to catastrophic events, as well as whether the near-average function is maintained [7]. Accordingly, adverse experiences range from single adverse life experiences—such as exposure to war—to aggregates across multiple negative events. However, there is no objective unitary index for evaluating the severity of adversity, because it is subjectively assessed by the observer’s cognitive system. The biopsychosocial model of challenges and threats explains individual differences in the experience of stress by focusing on the person’s appraisal of available resources [18]. The situation is perceived as threatening when the demands exceed the available resources. However, the stressor is evaluated as challenging, not threatening, when one perceives that they have sufficient resources to meet contextual demands [18]. For instance, Major et al. (1998) found that women who had more resilient personality resources to draw on (i.e., self-esteem, perceived control, and optimism) were less likely to appraise their upcoming abortions as stressful [19]. As such, the use of subjective ratings of the severity of adverse experiences may represent how the individual views the event. Hence, a cognitively based evaluation of the perceived adverse experience significantly shapes the trajectories of resilience toward adaptation, including the identification of obstacles as opportunities to be transformed, or undue appraisal of dangers as insurmountable barriers (see the cognitive appraisal parts of Figure 1).

### 3.2. Stressor Severity: The Importance of Cognitive–Emotional Process and Perception of Pain

However, while the severity of adversity is cognitively appraised, not all adverse experiences are equivalent in severity [20]. As mentioned above, the severity of the adversity can be modulated by the subjective expectation of perceived adverse event and its link to the consequential physical-pain feelings of emotional arousal (i.e., physical pain as a response to an emotion-regulation strategy [18]) (see the upper right box of Figure 1). Close related among these processes can link a subjective evaluation in cognitive appraisal system with emotional processing [21] (see the upper left box of Figure 1). For example, some adverse events are foreseeable in a given situation in the near future, such as life expectancy/survival rate of long-term illness; others are unexpected in a very short timeframe or in a certain period of the life trajectory. This unexpectedness of the adverse experiences can exert surprise (i.e., when something happens that is unexpected), which affects emotional regulation by amplifying awareness of physical-pain perception. The theoretical basis of surprise rests on predictions about sensations, and these depend on an internal generative model of the world [22]. Instead of just taking in information passively from the bottom-up, brains can make top-down hypotheses about the world to be tested against observations. They can then examine any incoming information that arrives, to see whether it fits that expectation or not (see [22] for detailed explanation). For instance, a surprising pleasurable event can produce unusually strong positive feelings, much stronger than an expected pleasurable event. Likewise, a surprising negative outcome can be peculiarly vexing or painful, more so than if the same outcome was to be expected [23]. As surprise increased, pleasure and pain were amplified, although the pain of doing worse than expected increased faster than the pleasure of doing better [24]. While surprise/unexpectedness can be a result of the conflict between cognitive and emotional anticipation, other conflicts between cognitive processing and somatic sensory response may result in physical-pain perception (see the upper right box of Figure 1). To illustrate, a mismatch between prior expectations and reality is referred to as a prediction error [22]. It is this incompatibility (or prediction error) that is the source of some emotional distress. In other words, humans intuitively attempt to strike an optimal balance between cognitive predictability and surprise. The mental effort required to cope with unexpected adverse experiences is accompanied by the product of this incompatibility (physical pain). These accounts likely offer a unifying model of pain perception, cognitive control, and emotional regulation. They illuminate the functional role of attention [25] in modulating these processes and neatly capture the special contribution of cortical processing to adaptive success. As such, how an individual appraises the adverse experiences in the first place can have a profound impact on how well people adapt to subsequent events. In line with this view, the cognitive appraisal may thus hold the key to our understanding of how likely people are to make it through an adverse event (i.e., positive adaption) [26].

### 3.3. The Role of Cognitive Appraisal in Cognitive–Emotional Processes

Multifaceted cognitive appraisal has manifested in many theories related to cognitive control and emotional regulation. For example, a multifaceted cognitive appraisal has been proposed as a way to explain responses to stressful events [27,28] or how different appraisals influence which emotions are experienced [29]. The theories relevant to an adverse event include one’s life and cognitive-based appraisal of the emotional encounter, and the linking of cognitive control and emotional regulation in cognitive appraisal. Recently, theoretical models [30,31] using a simple mathematical expression were proposed [31] to describe the important role of cognitive appraisal in explaining the mechanism of resilience; however, these expressions often lack evidence to support the neurocognitive findings underlying their elucidation. Some promising theories in the fields of cognitive and computational neuroscience have shown alternative insights. These theories conjecture that the machinery of the brain function is an internal generative model, involving a predictive mechanism that constantly monitors information between real-world sensations and the representation of its cognitively generated predictions [22,32,33].

Cognitive control and emotional regulation are two distinct processes in the process of emotional generation [34]. They are somewhat intertwined and typically modulated by the brain mechanism, including both top-down (goal-directed) cognitive control and bottom-up stimuli-driven attention processing [17]. According to this view, we see the resilience process as a top-down (goal-directed) and bottom-up (stimuli-driven) system involving an inner inference machinery mechanism that minimizes the surprise associated with sensory exchanges with the world [35]. Specifically, the brain compares its predictions with the actual sensory input it receives. The brain can “explain away” differences or prediction errors by using its internal models to determine the likely causes of the discrepancies. You can think of resilience as a recovery process in order to maintain optimal predictions (i.e., reducing prediction errors) between a perceived adverse event (e.g., prediction, anticipation, expectation) and sensory input (i.e., the severity of the adverse event, causing a sensation of physical pain). Recent evidence has demonstrated the commonalities of neural mechanisms connecting the experience of physical, emotionally aroused, and social-related pain, suggesting a substantial overlap between these phenomena [36,37,38,39,40]. Although pain perception is certainly related to physiological processes, how individuals react to a new episode of pain is shaped and influenced by previous experience, which is shaped by both cognitive [41] and emotional [40] influences on the perception of pain through sensory signals, constituting a complex emotional experience that varies significantly from one individual to the next. These data further connect pain perception as a consequence of the behavior and the internal state of the sufferer. These findings suggest that an appraisal causes one to experience emotions altering the subjective feelings of physical perceptions that produce pain (pleasure-maximizing) or relieve pain (pain-minimizing) [40,42].

From a neurocognitive perspective, the observer may alleviate subjective physical pain by reducing afferent nociceptive signals to the brain and descending modulatory systems that are activated endogenously by cognitive and emotional factors [43]. Therefore, in our view (see Figure 1), the positive adaptive function can be modulated by how the severity of adversities, their unexpectedness, and the subjective degree of the physical-pain apperception/painfulness resulting from adverse experiences are cognitively appraised. These variables can vary the brain mechanisms from top-down cognitive control to bottom-up stimuli-driven processing.

### 3.4. Cognition–Emotion–Perception Processes Connect Cognitive Appraisal

Experiencing adverse life events is a common factor in the development of mental health disorders [44]. However, not all individuals who encounter adverse experiences develop mental health issues; thus, there is considerable interest in understanding what makes an individual vulnerable or resilient to the deleterious effects of adverse experiences. Genetic factors doubtlessly play a role [45], but aspects of the adverse experience and complex cognitive factors regarding how the individual appraises or views that experience have also been argued to be key [44]. We reiterate that the cognitive appraisal of cognition, emotion, and perception is an important aspect of the stress process when undergoing adverse experiences [6,46,47,48]. Thus, people who demonstrate resilience appraise emotions as facilitative to one’s functioning.

For instance, the advantages of better emotional regulation in coping with adversity among the elderly include greater adaptation of the cardiovascular and immune system; greater cognitive resources, including a sense of self-efficacy; increased ability to seek social support; increased capacity to adapt to the intensity of stressful events; greater cognitive and affective integration; more mature defense mechanisms; low neuroticism (personality trait related to symptoms of depression, anxiety, and unhappiness); greater awareness; the use of proactive coping strategies; and greater satisfaction with life [49,50,51].

A study by Verduyn and colleagues (2011) explored the factors that affect the duration of emotional experience [52]. One aspect of the research focuses on the difference between rumination versus reappraisal of an emotional event. They explored how this difference affects the “duration” of an emotional experience and in which direction it proceeds (shortening or lengthening). The authors argue that cognition is the primary factor impacting duration and the experience of emotion by claiming that “thoughts appear to act as the fuel that stirs up the emotional fire and leads to a prolongation of the episode”. This study demonstrates the significance of cognitive appraisal by indicating its role in the duration of an emotional experience. Thus, it can be relevant to real-world applications in how individuals deal with adverse experiences. Hence, how we initially appraise the emotion-eliciting experience may lead to a prolongation of the episode/experience. The duration of the emotion is referred to as a part of the resilience process (see the upper parts of Figure 1). This concept alludes to the significance of congruence among emotions, appraisal, and cognition.

While most resilience theories are specific to particular populations (e.g., spousal loss, unemployment, and maltreated children) [53,54], there is a consensus call for a generic theory [55] that can be applied and generalized to different groups of people and potential adverse experiences [56]. One such theory commonly cited in the resilience literature [6,57,58,59,60,61] is the metatheory of resilience [55,62]. Richardson (2002) states that reactions to change or adversity are due to some protective or risk factors that influence one of four types of resilient outcome. This theory (and its accompanying model) is discussed because it can be potentially applied to different types of stressors, adversities, and life events. It can be used at various levels of analysis, such as individual, familial, and community, and includes a range of theoretical ideas from physics, psychology, and medicine. This theory provides a model of resilience that includes both trait and contextual factors. However, the model does not explain how cognition and emotion affect the reintegration process [63]. Hence, this metatheory has a significant conceptual drawback that diverts researchers’ attention from examining and understanding human resilience.

Our conceptual model (see Figure 2) underscores the importance of cognition–emotion and pain perception as processes underlying cognitive appraisal in modulating the resilience process. These top-down and bottom-up processes were put into the context of previous studies on resilience, such as the role of resource availability in resilience. An empirical example of resilience as a process can be found in the literature [64] of Montpetit, Bergeman, Deboeck, Tiberio, and Boker (2010). These authors explored the daily process of using resilient coping strategies among 42 participants aged 65 to 92 years, using a 56-day daily diary, with the expectation of one questionnaire completed per day. Their sample included 83% women; 54% of participants reported living alone, and 37% reported being married. Using multilevel-modeling statistical procedures, these authors found that, not surprisingly, better social support was related to greater use of resilient coping strategies (see the upper left box of Figure 1). We concur that some resources may significantly mitigate the subjective pain rating, i.e., accessibility to health-care facilities may ameliorate certain kinds of emotional and physical relief.

In the following section, we review recent brain-imaging studies from cognitive neuroscience and reason a distinct process in the brain circuitry that supports our conjecture on the dual processes’ reciprocal influence on human resilience. We tentatively propose a conceptual model that describes possible brain mechanisms for human resilience (see Figure 2) and synthesize findings from the neurocognitive perspective. This is further characterized by cognitive control/emotional regulation, and physical-pain perception processes are the dual routes that underlie human resilience. These arguments are supported by recent brain-imaging studies on resilience and show potential directions that differentiate the potential mechanisms of human resilience.

In this model, we dissect the two cognitive processes that arise from brain frontal networks (Figure 2). We propose that a disrupted brain frontal network results in impaired cognitive flexibility, leading to maladaptive behavioral outcomes. Top-down cognitive control modulates an individual’s cognitive function and emotional regulation. This helps to shift attention from the physical-pain perception induced by perceived adverse events via cognitive flexibility. However, if this network is disrupted, then it impairs the shifting efficiency, resulting in malfunctions of cognitive flexibility. This leads to individuals with excessive stress on the loop of physical-pain perception, without top-down cognitive modulation.

## 4. The Role of the Prefrontal Cortex in Cognitive Control, Emotional Regulation, and Pain Perception

Conceptualizing resilience in cognitive science has unique potential. Until the past decade, empirical studies of resilience predominantly focused on behavioral and psychosocial correlates of, and contributors to, this phenomenon. Recent studies have begun to examine neurobiological or genetic correlates of and contributors to resilience [45,65,66,67]. Technological advances in brain imaging and in measuring other biological aspects of behavior have made it more feasible to begin conducting research on pathways to resilient functioning from a multilevel perspective. Of these, brain-imaging tools are the most needed to determine the structural and functional networks that mediate the resilience process and its relation to cognition [68]. The cognition arise from the brain’s complex connectivity naturally requires sophisticated modeling approaches on a large variety of scales; the spectrum ranges from single-neuron dynamics over the behavior of groups of neurons, to neuronal network activities at the complex system level [69,70,71]. Thus, the connection between the microscopic scale (single-neuron activity) and macroscopic behavior (emergent behavior of the collective dynamics) and vice versa is key to understanding the brain in its complexity. While many conceptual models for explaining resilience have been proposed [29,30,72], none of them have connected findings from brain-imaging studies and synthesis with theories and concepts from cognitive neuroscience in order to explain how disrupted brain networks and cognitive dissonance lead to maladaptive behavioral outcomes. In this section, we discuss the neurocognitive perspective of the role of the frontal network in relation to human resilience (see Figure 2).

The architectonic organization of the prefrontal cortex is reflected in the five major frontal-subcortical circuits. These circuits mediate wide ranges of high-order cognition, such as executive functions [73] and personality changes [74]. To date, several emergent reviews have summarized studies of resilience in animal models, suggesting that certain brain regions are involved in perceiving adverse events. These brain regions include the medial prefrontal cortex (mPFC) [75], anterior cingulate cortex (ACC) [76,77,78], amygdala [79], and hippocampus [80]. For example, the abnormal function in specific regions, such as the amygdala, prefrontal cortex, anterior cingulate cortex, hippocampus, and parahippocampus, which are known to be related to anxiety and anxiety-related memory in individuals with PTSD, was discovered by using various brain-imaging tools. Particularly, many studies have reported altered amygdala and frontal activation in PTSD [81,82,83]. Shin et al. reported that PTSD patients showed amygdala hyperactivation and frontal hypoactivation and found that these regional activities were significantly correlated with the Clinician-Administered PTSD Scale score [83]. Indeed, some functional magnetic resonance imaging (fMRI) studies have shown an altered resting-state FC in some brain regions, including the amygdala, anterior cingulate cortex (ACC), and medial prefrontal cortex in patients with PTSD, as compared to healthy controls [84,85]. These findings suggest that disrupted frontal network connectivity may be related to those who struggled to recover from adverse experiences [86].

These brain regions form the brain reward circuitry and modulate neuroendocrine systems that underlie behavioral responses to stress-related adverse experiences [87,88]. Averse experiences are often turned into unwanted memories. Simple reminders can then trigger the involuntary retrieval of these memories. Milad and Quirk (2002) used animal models to show the role of mPFC in storing long-term memories for fear of extinction [89]. The mPFC is also part of a core network that not only supports the recollection of past episodes, but also imagines prospective events (e.g., when and where to meet a person). Thus, it appears that the dysfunction (higher effort switching cost at the same performance outcome) of mPFC between the recollection of past events and envisioning future events may hinder the positive adaption process. This can lead to a negative emotional response and, eventually, to developmental issues. Prior evidence [90,91] has shown that we can intentionally suppress the retrieval process to prevent unwanted memories from entering awareness by activating brain regions among the mPFC, amygdala, and hippocampus [92]. Of these regions, the mPFC was previously thought to be a hub that coordinates whole-brain connectivity. This is a consistent finding in human and animal reports and suggests that mPFC plays a crucial role in regulating cognitive appraisal to promote resilience in the face of adverse experiences (see Figure 2). For instance, a study using task-related fMRI examined brain responses after deliberate emotional regulation in trauma-exposed women with and without post-traumatic stress disorder (PTSD) [93]. The results showed that trauma-exposed non-PTSD subjects activated mPFC more after enhanced instruction than the PTSD group. They tended to activate these regions somewhat more than the non-traumatized control subjects, suggesting that mPFC may be a protective factor in the face of adverse event exposure and may be associated with resilience. Another study [94] showed that people who developed PTSD after experiencing adverse events often had brain cortical volume changes—specifically in mPFC, the amygdala, and the hippocampus. These imaging data suggest that these regions may regulate people’s ability to adapt. We propose that a disrupted brain frontal network results in impaired cognitive flexibility, leading to maladaptive behavioral outcomes.

Moreover, brain regions involved in the ACC, amygdala, and hippocampus are typically related to cognitive control and emotional regulation (see blue lines in the middle part of Figure 2). Cognitive control and emotional regulation complement each other and play an essential role in cognitive appraisal [95,96,97,98] (see Figure 1). For instance, studies have focused on functional correlates of these regions and found that they reflect an optimistic attitude [99,100], i.e., an individual is prone to adopting coping strategies with positive expectations regarding future events, resulting in proper emotional regulation (see upper parts of Figure 2). Moreover, these regions were previously thought to be responsible for cognitive control and emotional regulation [77,101], particularly the modulation of connections within frontal networks. For example, the mPFC projects to other stress-responsive structures such as the dorsal/caudal regions of the ACC; these are involved in the appraisal and expression of negative emotion, and the ventral–rostral portions of the ACC have a regulatory role with respect to limbic regions [88], generating emotional responses [77]. Moreover, these regions, specifically the ACC, are collectively responsible for survival-relevant goals such as physical pain [102] and negative emotion [101,103]. Other regions, such as the amygdala [79] and hippocampus [80], together with ACC, form networks that link to negative emotional processing [93] (see blue lines in the middle of Figure 2); this processing can minimize fear [79,104]. The above study highlighted that reappraisal—appraising the emotional situation—can act as an adaptive strategy to deal with hardship. Supporting evidence from neural correlates of reappraisal was also shown to increase activation of the lateral and medial prefrontal regions and decrease activation of the amygdala and medial orbitofrontal cortex. These findings support the hypothesis that the prefrontal cortex is involved in constructing reappraisal strategies that can modulate activity in multiple cognition–emotion processing systems [105].

Not only is this circuitry related to appraisal and emotional arousal, but these regions also overlap with the neural processes that distinguish affective from sensory pain dimensions, link emotion and pain, and generate central nervous system pain sensitization [40]. For example, a recent review [98] provides a comprehensive analysis of the influence of cognitive processes on pain perception and its potential integration of the contribution of attention, expectation, and reappraisal into the perception of pain. The authors highlighted the engagement of the ventrolateral prefrontal cortex during more complex modulation, leading to a change or reappraisal of the emotional significance of pain. Specifically, activation and functional connectivity between the descending pain control system, comprising the dorsolateral prefrontal cortex and the rostral ACC, positively correlates with pain relief, reduces activation of other pain-related regions of the brain, and is thought to also play a role in cognitive manipulations such as distraction, all of which contribute to pain reduction [40,102,106]. These commonalities of neural pathways connect the experience of physical pain, emotional feeling, and cognitive–evaluative thought, suggesting a substantial overlap between these phenomena. These functions are crucial and are coordinated by the mPFC, along with other brain regions, as seen in brain-imaging studies on resilience [101,103] (see the middle part of Figure 2).

These regions are also important to brain circuitry for the development of recovery from mental disorders [107]. As evidence converges in these brain regions, we highlight possible brain circuitry underlying resilience that likely exists: cognitive–emotional and physical-pain perception processes. These routes are seen in recent brain-imaging studies on resilience and show potential routes that distinguish the two forms of resilience. In particular, key brain regions (i.e., mPFC/ACC, the amygdala, and the hippocampus) act as provincial hubs that form a prefrontal network-modulated resilience process elicited by cognitive control, emotional regulation, and pain perception. To successfully adapt to adverse experiences, both cognitive–emotional and physical-pain perception processes are required to evenly distribute brain resources that switch between top-down cognitive-emotional adjustment and bottom-up physical-pain management (see Figure 2). Dynamic integration between both processes can form resilience processes. Hence, the capacity to shift or switch one’s perspective between representations of cognitive processes in response to a life change seems to be key. In addition, overlapped brain networks involving cognitive control, emotional regulation, physical-pain perception, and cognitive functions suggest that cognitive flexibility is critical to maintaining and coordinating various brain functions for adaptive behavior when facing adverse experiences (see Figure 2). In the following section, we explore the role of cognitive flexibility in two processes that underlie human resilience, as well as potential approaches to elucidate this mechanism.

## 5. Linking the Malleable Brain, Resilient Mind, and Adaptive Behavior: The Role of Cognitive Flexibility in Cognitive Control, Emotional Regulation, and Pain Perception

From a neurocognitive perspective, the brain regions mentioned above overlap with typical executive control networks related to task measurement and attention-shifting during different cognitive tasks, i.e., cognitive flexibility (see the upper parts of Figure 2). Recently, the ability to flexibly switch between strategies and cognitive frameworks has been proposed as a resilience factor [108]. In support of this notion, studies on resilience have shown an association between the individual’s cognitive flexibility in his or her neurochemical stress response systems and the neural circuitry involved in stress responses [109,110]. For instance, the role of mPFC is responsible for attention-shifting between different task representations [111,112,113]. It is likely that these regions exert an antagonistic effect on these two processes to modulate dose–response relationships for cognitive–emotional regulation and physical-pain perception [114]. To illustrate, previous studies have shown that children who experience a longer duration of trauma experience greater difficulties in attention-shifting [115], highlighting the link between cognitive flexibility and early-life adverse experiences. Therefore, the capacity to shift one’s perspective between different cognitive operations in response to a change in life seems to be important. However, studies [108,116] relating cognitive flexibility to resilience did not provide concrete behavioral measurements to examine the direct association between cognitive flexibility and resilience.

The mechanistic explanation of our conceptual model may be useful for future experimental design. From a neurocognitive perspective, cognitive flexibility can be measured by a task-switching paradigm, which measures the ability to shift attention between task sets [109]. Therefore, we speculate that maladaptive cognitive flexibility exhibits larger switching costs during attention shifts. This is seen in people who inadequately allocate their attention resources between cognitive control–emotional regulation and physical-pain perception processes, as well as in people who overly distribute attention resources to subjective physical-pain perception, overriding the regulating role of the top-down cognitive control process. By examining task-switching ability and its link with the duration of normal functioning, we can provide useful insights into how they view encountered adverse experiences. For example, a manipulated social stressor (e.g., public speaking) may become either a positive or negative experience [109]. These social stressors have been shown to be associated with enhanced physiological responses (cortisol reactivity) to a laboratory stress task [87,110] that may impair the ability to filter currently irrelevant task instruction information and reduce the flexibility needed to adapt behavior to task demands. A study [109] examined the effects of an acute psychosocial stressor (the Trier Social Stress Test) on a specific form of cognitive flexibility, namely that of set-shifting, which was assessed by the Berg’s Card Sorting Task (BCST). Their results showed that exposure to an acute social stressor promoted better performance on the BSCT, but this enhancing effect was minimized among individuals who appraised the stressor as being uncontrollable. These findings suggested the ability to effectively disengage from no-longer-relevant information, in favor of that which is newly relevant, i.e., cognitive flexibility, could have facilitated the effect of exposure to an acute social-stressor task.

It is likely that adverse experiences may disrupt attention shifts and lead to dysfunctions in cognitive flexibility between cognition–emotion regulations of pain perception. As reviewed above, larger switching costs usually demand more effortful, top-down, high-level cognitive control and emotional regulation to balance the over-excited, bottom-up, stimuli-driven attention to physical-pain perception. Here, we tentatively hypothesize that individuals with low resilience may experience recurring physical pain, regardless of the trigger, due to a subjective physical-pain-perception loop. Poor modulation of cognitive flexibility, which mediates cognitive–emotion processing and physical-pain-perception processing, weakens the top-down process and revives the loop of physical pain.

Moreover, from a brain-network-organization perspective, the reorganization of brain networks and their association with adverse experiences may serve as an important predictive marker for normal functioning [117]. Some people experience higher levels of severe adversity; this can disrupt the brain network [86] and weaken the cognitive flexibility needed to adjust negative emotions back to their normal affective state. Worse still, an adverse event can lead to a large surprise that may impair the cognition–emotion processing and amplify larger switching costs for cognitive flexibility. Unexpected events can cause an automatic interruption of ongoing mental processes, followed by an attentional shift and attentional binding to the events. These are often inadequate to resume proper functioning [118]. For instance, frequent shifts in different cognitive representations may exhaust or distract the brain resources and thus disrupt self-regulatory or self-efficacy. This can lead the person to have low resilience and a failure to resume daily functioning. These examples and evidences resonate with our conceptual model and likely imply a disrupted brain network, leading to surprise or unaccepted loss in encountered adverse events. This leads to dysfunction and avoidance of negative outcomes; this can also lead to impaired cognitive flexibility.

Furthermore, these adverse experiences may be accompanied by the dysfunction of cognitive functions resulting from brain injuries [119], such as abused and neglected children, victims of violent crime and assault, refugees, veterans, and minority groups. To support this notion, several studies have established a link between cognitive functions and traumatic brain injury [120,121]. Analogous to the neural plasticity that occurs in response to brain injury [10,122,123,124], resilience can be viewed as the developmental process of an individual adapting after exposure to adverse events or traumatic experiences [125]. According to this view, adversity is thought to exert a damaging effect on one or more neural substrates, and mechanisms of neural plasticity can lead to recovery. This might lead to the conclusion that certain individuals, classified as resilient, carry an increased innate capacity (i.e., the efficiency of neural plasticity), above and beyond normative levels, to recover from adverse experiences that affect the brain. These new perspectives on neural plasticity make it possible to undertake empirical studies on the relationship between neural plasticity and resilience. This can, perhaps, enable an examination of the direct link between these two adaption processes.

## 6. Future Perspectives and Concluding Remarks

This manuscript provides evidence to support the relevance of the brain and cognition in studying resilience (Table 1). The goal of this paper is to propose a testable working model of human resilience via cognitive-based experimental approaches. Despite the fact that some of the critical concepts objectively detailed here remain challenging, we cautiously offer some promising clues, questions that remain, and possible experimental methods to test this model (Table 2). In providing this conceptual framework, inadequate though it may be, we hope to spur further discussion about the nature of contemplative practice and how the neurocognitive perspective of psychological resilience may help us better understand the causes and conditions of human response to adverse experiences. These examinations may help to answer how the individual’s cognitive appraisal of the adverse experience influences the recovery trajectory, as well as the ability to switch between different task sets, reflecting the efficiency of attentional shifting among cognitive control, emotion regulation, and pain perception. With the advancements in brain imaging of the cognitive process, we can link the subjective evaluated severity of adverse experience and the likelihood of occurrence with the recovering outcome of normal functioning. By examining flexibility functioning and its link to appraised adverse experiences, we may be able to uncover how different degrees of flexibility of cognitive control, emotion regulation, and pain perception influence coping strategies. For instance, a return to normal functioning may be sufficient within the context of severe adversity [7]. This indicator can be easily assessed in domain-specific performance: an indicator for schoolchildren might be an academic achievement. A more appropriate indicator for military personnel would be the absence of psychiatric symptoms. Enriching these results and insights to understand the circumstances of adversity are studies focusing on specific types of adverse experiences. Moreover, some abstract concepts with subjective evaluation or feelings might need more rigorous validity and reliability. For instance, cognitive appraisal can be assessed by coping [126,127,128] and cognitive styles [129]. Cognition–emotion and its flexibility can be assessed through either a validated questionnaire [130,131] or psychological tasks [132,133]. Furthermore, the mismatch between prior expectation and reality (i.e., surprise) can be characterized by a specific pattern of electrical activity [134] or brain activation [35,135,136].

In sum, we reviewed the current literature of resilience research, suggesting that resilience is a process of experienced adverse events and positive adaption (i.e., avoidance of negative outcome). We also underscore the role of cognitive appraisal in perceiving the severity of adverse experiences that shape an individual’s resilience trajectory, and the conflict between ongoing actual experiences and the brain’s modeled expectations. These can induce emotional distress and the physical consequences of painful feelings. Cognitive flexibility thus plays a primary role in modulating mismatch difference between expected and real-world scenarios, through the regulation of cognitive control of emotions and pain perception. These conjectures have received support from brain-imaging studies on PTSD and those who experienced disasters [57]. Importantly, we argue that the brain’s frontal network plays an important role in regulating cognitive control/emotional regulation processes and the physical-pain perception process: these processes can be modulated by top-down (goal-directed) and bottom-up (stimuli-driven) attention strategies, respectively.

Overlapping brain networks, including cognitive control, emotional regulation, physical-pain perceptional, and cognitive functions, suggest that cognitive flexibility plays a role in maintaining and coordinating various brain functions for adaptive behavior when facing adverse events. The dysfunction of cognitive flexibility begins an infinite loop of failed top-down regulation to keep up the normal functioning of the resilience process. We believe that the continued study of resilient trajectories carries substantial potential for ongoing refinements of existing theories of normal human development. Moreover, a longitudinal examination of resilience through quantitative, qualitative, and experience sampling methodologies will enhance our understanding of this construct. We further encourage the use of measures and theories with an intersectional lens.

## Figures and Tables

**Figure 1 ijerph-16-05123-f001:**
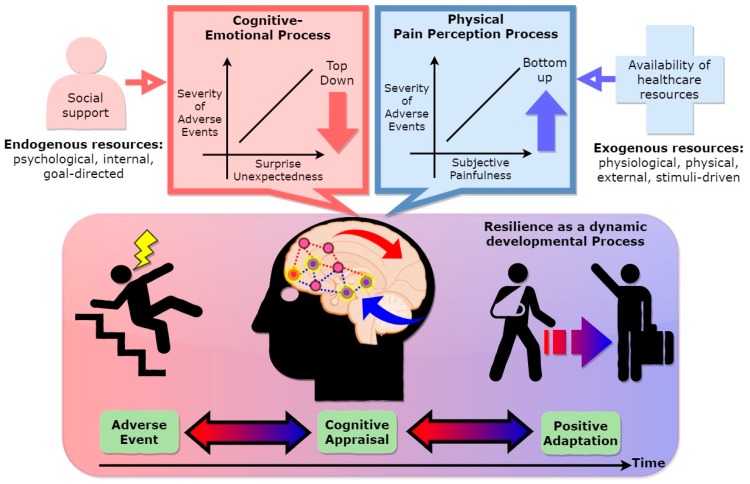
Cognitive appraisal of resilience (CAR) model.

**Figure 2 ijerph-16-05123-f002:**
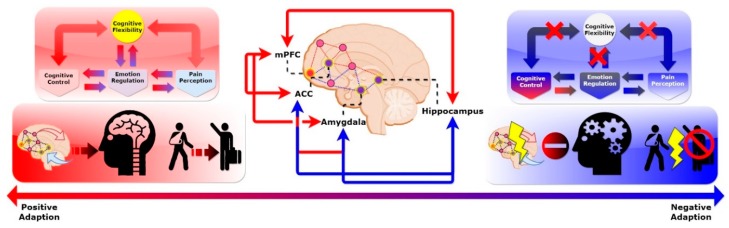
Two conceptual processes within the CAR model.

**Table 1 ijerph-16-05123-t001:** Overview of the key literature for each section.

	Section Highlights	Authors (Year)	Article (Title)	Source (Journal)	Relevant Findings and Their Implications to the Highlights
Section 1	Human resilience may protect us from developing mental health issues.	Southwick and Charney (2012)	The Science of Resilience: Implications for the Prevention and Treatment of Depression	Science	This paper shows the importance of resilience in protecting humans from developing mental issues. Authors review interdisciplinary factors that influences resilience and call for integrating salient concepts of resilience into relevant fields of medicine, mental health, and science.
		Southwick et al. (2014)	Resilience Definitions, Theory, and Challenges: Interdisci-plinary Perspectives	European Journal of Psychotrau-matology	This paper summarizes the discussion from multidisciplinary experts in the study of psychological resilience for the most pressing current questions in the field of resilience research. The consensus among these experts was resilience is a complex construct and multiple levels of analysis from interdisciplinary perspectives are urgent and needed.
Section 2	Resilience refers to a dynamic process encompassing positive adaptation within the context of significant adversity.	Fletcher and Sarkar (2013)	Psychological Resilience: A Review and Critique of Definitions, Concepts, and Theory	European Psychologist.	In this paper, the authors review the literature and operationalized definitions of resilience, suggesting positive adaptation must be conceptually appropriate to the adversity examined. They argue more conceptual study is needed that should take into account the multiple demands individuals encounter, the meta-cognitive and -emotive processes that affect the resilience–stress relationship, and the conceptual distinction between resilience and coping.
		Mancini and Bonanno (2009)	Predictors and Parameters of Resilience to Loss: Toward an Individual Differences Model	Journal of Personality	In this paper, Bonanno, et al. discussed what is the resilient capacity by reviewing prior work (G.A. Bonanno, 2004). They provide an operational definition of resilience as a specific trajectory of psychological outcome and describe how the resilient trajectory differs from other trajectories of response to loss. They integrate these individual differences in a hypothesized model of resilience, focusing on their role in appraisal processes and the use of social resources. In line with this paper, we suggest the period between experience difficulties completing their normal tasks at daily functioning until begin to return to their pre-loss level of functioning that able to complete tasks in daily life may see as indexes of positive adaptation.
Section 3	Cognitive appraisal mediates cognition–emotion–perception processes from adverse experience to positive adaptation.	Major et al. (1998)	Personal Resilience, Cognitive Appraisals, and Coping: An Integrative Model of Adjustment to Abortion	Journal of Personality and Social Psychology	In this paper, the authors found that women who had more resilient personality resources to draw on (i.e., self-esteem, perceived control, and optimism) were less likely to appraise their upcoming abortions as stressful. The results implied the role of cognitive appraisals in personal resilience and coping are discussed as possible mediators of this relationship.
		Verduyn et al. (2011)	The Relation Between Event Processing and the Duration of Emotional Experience	Emotion	In this paper, the authors explored the factors that affect the duration of emotional experience. Specifically, they focus on the difference between rumination versus reappraisal of an emotional event. They explored how this difference affects the “duration” of an emotional experience, and in which direction it proceeds (shortening or lengthening). The authors argue that cognition is the primary factor impacting duration and the experience of emotion, by claiming that “thoughts appear to act as the fuel that stirs up the emotional fire and leads to a prolongation of the episode”. This study demonstrates the significance of cognitive appraisal by indicating its role in the duration of an emotional experience.
		Doukas et al. (2019)	Hurts So Good: Pain as an Emotion Regulation Strategy.	Emotion	In this paper, the authors tested two primary hypotheses: some people will choose to inflict pain to regulate negative emotional states, and (b) pain provides effective short-term relief from negative emotion. Their results suggest physical pain as a response to an emotion regulation strategy, linking physical pain, emotional response, and cognitive appraisal.
		Eysenck et al. (2007)	Anxiety and Cognitive Performance: Attentional Control Theory.	Emotion	In this paper, Eysenck et al. discussed top-down and bottom-up processing in regulating two central executive functions related to attentional control: inhibition and shifting. Mental health impaired (e.g., anxiety) disrupts these two functions by weakens the degree to which inhibitory mechanisms can regulate automatic responses, that is, anxiety weakens top-down cognitive control. They illuminate the functional role of attention in modulating these processes and neatly capture the special contribution of cortical processing to adaptive success. This theory has a profound impact on mental health research. We speculate this regulatory mechanism also exists in the process of positive adaption in the face of adversity.
Section 4	Frontal brain network connects cognitive control, emotion regulation, and pain perception	Shin et al. (2004)	Regional Cerebral Blood Flow in the Amygdala and Medial Prefrontal Cortex during Traumatic Imagery in Male and Female Vietnam Veterans with PTSD	Archives of General Psychiatry	Shin et al. have reported that PTSD patients showed amygdala hyperactivation and frontal hypoactivation and found that these regional activities were significantly correlated with the Clinician-Administered PTSD Scale score. Indeed, some functional magnetic resonance imaging (fMRI) studies have shown an altered resting-state FC in some brain regions, including the amygdala, anterior cingulate cortex (ACC), and medial prefrontal cortex in patients with PTSD, as compared to healthy controls. These findings suggested that disrupted frontal network connectivity may be related to those who struggled to recover from adverse experiences.
		Milad and Quirk (2002)	Neurons in Medial Prefrontal Cortex Signal Memory for Fear Extinction	Nature	Aversive experience events are often turned into unwanted memories. Simple reminders can then trigger the involuntary retrieval of these memories. Milad and Quirk (2002) used animal models to show the role of mPFC in storing long-term memories for fear of extinction [89]. The mPFC is also part of a core network that not only supports the recollection of past episodes but also imagines prospective events (e.g., when and where to meet a person).
Section 5	Cognitive flexibility moderate resilience by regulating frontal brain circuitry.	Gabrys et al. (2017)	Traumatic Life Events in Relation to Cognitive Flexibility: Moderating Role of the BDNF Val66Met Gene Polymorphism	Frontiers in Behavioral Neuroscience	In this paper, the authors showed that children who experience a longer duration of trauma experience greater difficulties in attention-shifting, highlighting the link between cognitive flexibility and early-life adverse experiences.
		Gabrys et al. (2019)	Acute Stressor Effects on Cognitive Flexibility: Mediating Role of Stressor Appraisals and Cortisol.	Stress	In this paper, the authors examined the effects of an acute psychosocial stressor (the Trier Social Stress Test) on a specific form of cognitive flexibility, namely that of set-shifting, which was assessed by the Berg’s Card Sorting Task (BCST). Their results showed that exposure to an acute social stressor promoted better performance on the BSCT, but this enhancing effect was minimized among individuals who appraised the stressor as being uncontrollable.
		Bonanno and Burton (2013)	Regulatory Flexibility: An Individual Differences Perspective on Coping and Emotion Regulation	Perspectives on Psychological Science	In this paper, the author describes emotion-regulation flexibility, defined as the matching of emotion-regulation strategy to environmental circumstance. They segmented emotion-regulation flexibility into three separable valuation systems: (1) how we read the situation or context-sensitivity; (2) a repertoire of behaviors; and (3) the ability to regroup by using corrective feedback. This paper echoes our concept of flexibility in regulating resilience: We focus on neurocognitive aspects of flexibility and suggest its mechanism underlies human resilience.
Section 6	Examples, questions remain and future direction.	Parsons et al. (2016)	A Cognitive Model of Psychological Resilience	Journal of Experimental Psychopathology	The authors proposed a cognitive model to describe the role of selective information processing in positive adaption in the face of adversity. In their theoretical framework, they provide some possible psychological task manipulations (e.g., threat cues to induce anxiety) that may enable the study of the development of cognitive functions that are important in the resilience process. This paper resonates with our paper and shows the potential to examine whether the cognitive aspect plays a role in resilience.

**Table 2 ijerph-16-05123-t002:** Future directions and research questions.

1. Brain structures and functions investigation
What is the fundamental difference between brain structure and function in individuals who successfully return to normal functioning after the experienced adverse events? Are brain structure and function properties different in individuals who experienced different levels of the severity of adversity, and do they differ when experienced short-term versus long-term adversity? What aspects of brain structure and function differentiate individuals who function resiliently, despite experiencing early adversity, from those who function in a non-resilient fashion and encounter adversity early in life (i.e., what is the role of early experience?)? Are particular areas of the brain more likely to be activated in resilient than in non-resilient functioning during challenging or stressful tasks (manipulated adverse situation)?
2. Brain plasticity and cognitive training intervention
Would an individual who experienced long-term adversity alter its neural plasticity? If so, would cognitive-training intervention preserve the cognitive functioning that facilitates the resilient behavior? As our model suggested, mPFC plays an important role, so does delivering brain stimulation over mPFC boost individuals with low resilience? Is the efficiency of cognitive switching distinguishing low and high resilience individuals, and if the training intervention of cognitive flexibility training, can we make the low-resilience individual more resilient? Does long-term adversity change cognitive flexibility performance, and would it differ in high- versus low-resilience individuals? Does task-switching training make an individual with low resilience more resilient, and which brain region responds to the training? Is there a difference in the brain frontal network modularity between pretraining and post-training of cognitive flexibility and its relation to the performance of adapting to adversity?
3. Developmental consequences in resilience
Are there sensitive periods in which the capacity to return to normal functioning is possible, or is it possible to bounce back from adverse experience across the lifespan, and does it correlate with the development of cognitive flexibility? Do the neural data that act as a precursor to predicting the probability of return of normal functioning alter if circumstances change? Are the factors that lead to adult resilience similar to those found for children and adolescents, and whether they function cumulatively and interactively? Does neural plasticity play a role in the development and maintenance of resilient functioning, and does the efficiency of neural plasticity operate differently in individuals classified as resilient?

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
