# Peer review of "Neurocognitive Mechanism of Human Resilience: A Conceptual Framework and Empirical Review"

_ijerph, 2019, doi:10.3390/ijerph16245123_

Round 1

Reviewer 1 Report

This paper reviews current research assessing the cognitive mechanisms underlying resiliance. In particular, Yao and Hsieh focus on the effects of cognitive appraisal on recovery and efficiency to shift between cognitive control, emotional regulation and pain perception. The authors then relate these cognitive processes to brain structures and pathways and suggest a potential neurocognitive model of resilience. 

The paper is interesting and the neurocognition of resilience is novel and under-researched. The authors provide a sufficient overview of the cognitive literature however the scientific accuracy of the manuscript is sometimes questionable. For example, many conclusive statements are made without citations and it isn't clear whether these are the authors opinions or text that hasn't been cited correctly (e.g. see section 3.2).

The manuscript is often too wordy, resulting in unclear points. The entire manuscript should be written more concisely. Also, perhaps using tables to summarise current literature would be more helpful.

Other more minor suggestions include:

While I see how the three quotes relate to the paper, I don't see them as contributing anything.

The authors should define top down and bottom up processing at it's first mention rather than in the second last paragraph of the paper.

While the figures refer to physical pain, it is not clear whether text refers to physical pain and/or emotional pain. Is emotional pain considered? If not, this paper relates only to resilience following an injury or illness, and this needs to be specified throughout the entire manuscript. Clarification is needed and pain should be defined.

There are far too many references to the figures throughout the text. While reading, there's a mismatch in seeing the figures but not reading about parts of them until pages later. The authors should include a brief overview description of the figures.

Examples given don't always clearly illustrate the point e.g. line 398.

Reviewer 2 Report

This review pays attention to an important psychological issue, the resilience. It is my pleasure to read this review. I had some suggestions for this review.

There is some information was repeated from line 58- line 77. Please show the essential content and prevent repeat the same information.

I did not see an adequate literature review about the definition of resilience and how to assess resilience in previous studies. The review should cover the important tools for assessing resilience.

I could not see any rationale to support “We propose that a disrupted brain frontal network results in cognitive dissonance leading to maladaptive behavioral outcomes” (line 92) in this review.

In section 6, I could not see a clear rationale to directly support that cognitive flexibility contributes directly to resilience. For example, I could not see why the task-switching ability was related to emotional regulation in adverse events. It might be right, however, more reference should be provided to support the essential idea.

As the discussion in the review was complex, I will suggest having a table to show the summarized conclusion in the 7 sections of the review.

Round 2

Reviewer 2 Report

I have no further command.